# Microstructure and Formation Mechanism of Ultrasound-Assisted Transient Liquid Phase Bonded Magnesium Alloys with Ni Interlayer

**DOI:** 10.3390/ma12223732

**Published:** 2019-11-12

**Authors:** Yinan Li, Chengfei Yang, Zilong Peng, Zhiyuan Wu, Zhuang Cui

**Affiliations:** 1School of Mechanical and Automotive Engineering, Qingdao University of Technology, Qingdao 266000, China; 2Welding Engineering Program, The Ohio State University, Columbus, OH 43221, USA

**Keywords:** Mg alloy, ultrasound, transient liquid phase bonding, microstructure, mechanical performance, joining mechanism

## Abstract

Ultrasound-assisted transient liquid phase bonding (U-TLP) has been regarded as a promising brazing process to join magnesium alloys with a Sn and Zn interlayer; however, the formation of brittle magnesium intermetallic compounds (Mg_2_Sn, MgZn, and MgZn_2_) compromises the mechanical properties of the joints. In this study, Mg alloy U-TLP joints with a Ni interlayer were evaluated based on shear strength and hardness measurement. Microstructural evolution along with ultrasonic duration time and intermetallic compound formation were characterized using X-ray diffraction and electron microscopy methods. The results show that incremental ultrasonic durations of up to 30 s lead to the microstructural evolution from the Mg_2_Ni layer, eutectic compounds (Mg_2_Ni and α-Mg) to α-Mg (Ni), accompanied by shear strength increases. The maximum value of the shear strength is 107 MPa. The role that ultrasound vibration played in brazing was evaluated, and showed that the MgO film was broken by the acoustic softening effect when the interlayer and base metal were solid. As the MgO and Mg substrate have different stress reduction τ, this plastic mismatch helps to break the oxide film. Additionally, the diffusion between the solid Mg substrate and Ni interlayer is accelerated greatly by the acoustic pressure based on the DICTRA dynamic calculation.

## 1. Introduction

Magnesium and its alloys are potential candidates to replace steel and aluminum alloys in manufacturing industries in future, due to some of their unique properties like a low density, high strength-to-weight ratio, good castability, and high damping capacity [1,2]. Currently, various methods have been applied for welding magnesium alloys such as arc welding [3,4], laser welding [5,6], brazing [7], friction stir welding [8,9], and transient liquid phase bonding [10]. However, each method has its own limitation [11,12,13], such as the low penetration with arc welding, serious pore defect with laser welding, brittle intermetallic compounds (IMCs) formed with brazing, and difficulty of joining complicated structures with friction stir welding.

Ultrasound-assisted transient liquid phase bonding (U-TLP), which has been developed from the TLP method, is a low-cost, high-efficiency, green, and reliable joining method, which has been applied to join different kinds of alloys [14,15] or dissimilar alloys [16,17,18]. Xu et al. joined the Mg alloy within 1 s using a Zn layer via U-TLP; however, the joint was filled with MgZn and MgZn_2_, which led to a 42 MPa shear strength [19]. Lai et al. had to prolong the ultrasonic treatment time to 120 s in order to obtain a shear strength of 106 MPa, as the joint was mainly composed of α-Mg (Zn) [20]. Wang et al. adopted Ag-28Cu foil to join Mg alloys and the maximum shear strength was 92 MPa [21]. 

In this work, pure nickel foil was utilized to braze the Mg–Mn–Ce alloy within 30 s via U-TLP in air to get a high mechanical performance and non-defect joints as Mg–Ni intermetallic compounds (IMCs) have lower formation enthalpy and Gibbus energy, and the Ni foil has better oxidation and corrosion resistance compared with Zn, Al, and Sn foil [22,23,24]. The influence of the ultrasonic treatment time on variation of the microstructure of the brazing seam was studied. Then the mechanical performance of the joints was measured. In particular, the role that ultrasound vibration (USV) played during the U-TLP bonding was discussed in detail.

## 2. Experimental

The base metal (BM) used in this work was Mg–Mn–Ce alloy (1.3–2.2 wt % Mn, 0.15–0.35 wt % Ce, 0.2 wt % Al, 0.3 wt % Zn), which was provided by Northeast light alloy Co. Ltd. (Harbin, China). The BMs were cut into 20 × 20 × 3 mm^3^ sections. The pure Ni interlayer, which was 50 μm thick, was cut into 20 × 20 mm^2^ pieces. All of the BMs were sanded by SiC papers to 800 grit finish to keep the surface roughness as 12.5 μm, which helps to remove the oxide film during ultrasonic vibration. The BMs and Ni foils were cleaned by acetone using absorbent cotton and then ultrasonically cleaned in alcohol for 10 min. The schematic of the U-TLP apparatus and the arrangement of the sample are shown in Figure 1. The samples were heated by a medium frequency induction heating device with a 3 °C/s heating rate. The experiments were conducted in air with 0.15 MPa preload pressure. When heated to 520 °C, the samples were separately treated with ultrasonic vibration for 10 s, 15 s, 20 s, 25 s, 27 s, and 30 s. Then the ultrasonic vibration and heating were shut down and the samples were cooled down at the rate of 50 °C/min. The sonotrode was moved away and then the preload pressure was removed when the temperature of the samples decreased blow 200 °C. 

The microstructure of the joints was characterized by a scanning electron microscope (SEM, MERLIN Compact, Qingdao, China) coupled with an energy dispersive x-ray spectrometer (EDS). To ensure reliable strength results, three samples with a length and width of 20 × 10 mm for each ultrasonic treatment time were prepared for tensile tests. The shear strength was measured by an electromechanical test machine (WDW-50KN) with a loading speed of 0.1 mm/s. The fracture surfaces were characterized using x-ray diffraction (XRD, D/max 2200VPC) with Cukα radiation to identify eutectic phases. The tube voltage and current were 40 kV and 40 mA, respectively. 

The nano-hardness at the interface was measured by a nanoindentation instrument (MTS) with a testing depth of 500 nm and a dwelling time of 10 s. The Vickers hardness of the brazing seam was measured by an HXD-1000TM machine with an indentation load of 10 g and a holding time of 10 s.

## 3. Results

### 3.1. The Microstructure of Mg/Ni/Mg Joints

Figure 2 shows the microstructural changes of the joints with increasing ultrasonic treatment time and the joints clearly show three zones: a total reaction zone (L1); the remaining Ni interlayer zone (L2), and the diffusion and dissolution zone (L3 and L4). The EDS analysis of different regions in the joints is listed in Table 1. It can be seen from Figure 2a that a continuously distributed grey layer was formed at the interface between the Ni interlayer and the Mg BM with 10 s ultrasonic duration time. The EDS analysis showed that this layer (point A) was identified as Mg_2_Ni. The width of L3 (Mg_2_Ni layer) and L1 were 2.3 μm and 51.3 μm, respectively. The formation of the Mg_2_Ni means that the oxide film on the surface of the Mg was eliminated by USV within 10 s [18,20,25]. It was said in [14] that due to the acoustic plasticity effects, the BM and the oxide film were triggered in varying deformation and this plastic mismatch between the BM and the oxide film led to the breakage of the oxide film. So, the mutual diffusion between the Mg BM and the Ni interlayer proceeded and the Mg_2_Ni IMC layer was created after the fresh Mg and Ni atoms came into contact with each other.

At 15 s ultrasonic treatment time, the large blocky grey Mg_2_Ni (point B) and dark α-Mg were formed in L3 and L4. This is because the eutectic reaction, Equation (1), occurs at the interface
(1)$$L\overset{\;506\;{}^{\circ}C\;}{↔}{\mathrm{Mg}}_{2}\mathrm{Ni}\;+\;\alpha -\mathrm{Mg}(\mathrm{Ni}).$$

The width of L2 decreased to 37.8 μm and L1 increased to 66.2 μm. When the ultrasonic treatment time was 20 s, a large amount of the Ni interlayer was dissolved as the eutectic reaction proceeded. The width of L2 declined to 15.3 μm, while the width of L3 was up to 102.9 μm. When the ultrasonic treatment time was 27 s, the Ni interlayer was totally dissolved, L1 reached the maximum width value of 110.0 μm, and the joints were composed of the (Mg_2_Ni + α-Mg(Ni)) eutectic. Due to the intervention of the ultrasound field in this period [26,27], the dissolution and diffusion between the BM and the interlayer became faster, and the 50 μm thick Ni interlayer dissolved completely within 27 s. 

When the ultrasonic time extended to 30 s, it can be seen that the width of L1 dropped sharply to 18.1 μm. The joints mainly consisted of α-Mg(Ni) solid solutions with a small amount of the (Mg_2_Ni + α-Mg) eutectic. In this period from 27 s to 30 s, the synergy of the USV and sonotrode pressure accelerated the process of the liquid eutectic extrusion at the joint interface. 

Figure 3 shows the variation of width of zones L1–L4. It can be seen that L3 was wider than L4. This is mainly because the ultrasonic horn was placed on the L3 side of the BM and the ultrasonic wave had the characteristic that it can be attenuated in solid and liquid phase [28,29,30]. The L3 zone received USV that transferred across the solid BM while the L4 zone received USV that transferred across the solid BM, liquids in L3, and solid Ni interlayer, which meant that more energy from the USV is dissipated in the L3 zone than the L4 zone. Therefore, the degree of dissolution and diffusion in the L3 zone was higher than that in the L4 zone. 

Figure 4 shows the Mg and Ni atom distribution along lines in the joints with 20 s, 27 s, and 30 s ultrasonic treatment time. It shows that the content of Ni atoms in the 20 s joint was between 20–40 at % and its distribution width was around 102.9 μm; the content of Ni atoms in the 27 s joint reduced to less than 20 at % and its distribution width increased to 110.0 μm; the content of Ni atoms in the 30 s joint was less than 10 at % and its distribution width decreased to 18.1 μm. This distribution of Ni and Mg atoms indicates that USV promoted the dissolution and diffusion between the BM and the interlayer, which lead to the composition of the joints becoming homogenous with the extension of ultrasonic treatment time.

### 3.2. Mechanical Performance of the Mg/Ni/Mg Joints

Figure 5 shows the change of the shear strength of the BM and the joints under the incremental ultrasonic treatment time. Mg–Mn–Ce alloy is a kind of wrought magnesium alloy and it can be seen that some α-Mn particles precipitate along the grain side of α-Mg in the BM in as-received condition, while these precipitates were dissolved in the matrix after heating and USV [31,32]. The shear strength of the as-received BM was 105 MPa. The shear strength of the BM declined to 96 MPa after heating to 520 °C and 30 s of USV, which is 8.6% below the strength of the as-received BM. This reduction of shear strength is the result of the reinforced phase reduction and the ultrasonic softening caused by a hybrid field of thermal and USV [33,34,35].

As shown in Figure 5, with the increasing ultrasonic treatment time, the shear strength increased because of the ultrasonic softening and acoustic steaming effects. These two effects help to eliminate the oxide film on the BM and accelerate diffusion and dissolution between the BM and the Ni interlayer. However, prolonged ultrasonic treatment time up to 35 s induced the formation of unfilled zones caused by excess extrusion of reaction products, decreasing the shear strength of the joint. When the ultrasonic duration time was 10 s, the average shear strength of the joints was 19 MPa shown in Figure 6, which means that the oxide film was broken, and the bonding was able to form at the interface as the Mg_2_Ni layer was generated here (Figure 2a). When the ultrasonic treatment time was 20 s, the shear strength increased to 32 MPa. From Figure 7a it can be seen that the fracture originated in the Mg_2_Ni layer rather than the (Mg_2_Ni + α-Mg) eutectic and propagated along the interface between the Mg_2_Ni layer and the eutectic. According to the results of the Vickers hardness and nanoindentation hardness of the phase shown in Figure 6, the Vicker Hardness (HV) hardness of Mg_2_Ni was 142.0 HV and the nano-hardness of it ranged from 2.265 to 2.969 GPa; whereas the HV hardness of the eutectic was 130.2 HV and the nano-hardness of it ranged from 1.511 to 1.714 GPa. In other words, the Mg_2_Ni layer is harder than the (Mg_2_Ni + α-Mg) eutectic, which means that the Mg_2_Ni layer formed at the surface of Ni foil is the most brittle area in the joint. These harder regions are prone to be the crack initiation spot, which can be confirmed by the x-ray diffraction analysis of the fracture surfaces. Additionally, the pop-in events occurred in load–displacement curves shown in Figure 6b, which means that the phases underwent an elastic–plastic transition and then micro-deformations were induced [36,37]. 

After 27 s ultrasonic treatment, the Ni interlayer was dissolved completely and the joint was only composed of the Mg_2_Ni+α-Mg(Ni) eutectic, which meant that the shear strength increased to 80 MPa. It can be seen that the fracture path originated and propagated along the eutectic in Figure 7b. Without the existence of a continuous, distributed, brittle, Mg_2_Ni layer, the fracture path will be blocked by the α-Mg in the eutectic to some extent, which leads to the improvement of shear strength.

When ultrasonic time extended to 30 s, the joint was mainly composed of α-Mg(Ni) solutions and the shear strength reached its maximum value of 107 MPa, which is 102% of the as-received BM and 111% of the BM after heating and ultrasonic treating. This is mainly because the abundant brittle Mg_2_Ni IMCs were squeezed out and α-Mg (Ni) remained in the joint, with a Vickers hardness of 56.4 HV and nano-hardness ranging from 0.784 to 0.821 GPa. There were two fracture paths formed at this joint, shown in Figure 7c. One was located in the brazing seam, the other was at the Mg BM. The nano-hardness and Vickers hardness of this phase in the Mg BM were close to those of α-Mg (Ni), which means that the origin of the crack will occur randomly in either the brazing seam or the Mg BM when the joints are subjected to the same shear force. From Figure 2f it can be seen that a small amount of the eutectic still existed in the seam. As the shear force continued to increase, the fracture was propagated more quickly along the eutectic in the brazing seam than in the BM. Therefore, the joint was finally broken at the brazing seam. The fracture morphology of the 30 s joint shown in Figure 7d belonged to a typical brittle fracture feature. The large, flat α-Mg and small, fragmentized Mg_2_Ni were observed there. Jin et al. studied the joints using an Ni interlayer via TLP [38]. The samples were heated to 515 °C and kept at that temperature for 60 min, and the shear strength of them was 36 MPa. The main reason for this relatively low shear strength was that a large amount of the Mg_2_Ni existed in the brazing seam. With the U-TLP method, the brittle Mg_2_Ni IMCs were squeezed out and the joint was mainly composed of α-Mg [20,21]. 

When ultrasonic treatment time was 35 s, the shear strength dropped sharply and some defects like unfilled zones were generated in the brazing seam. The main reason was that the excessive reactive products were squeezed out under the action of the acoustic steaming effect.

Figure 8 shows the XRD patterns taken from the fracture surface of the joints with 20 s and 30 s ultrasonic treatment time. It shows that from the fracture surface of the 20 s joint the Ni had more intensive XRD pattern. This means that the fracture was located near the unreacted Ni interlayer, which was the place that the Mg_2_Ni layer was concentrated. In the fracture surface of the 30 s sample, the phases consisted of α-Mg and Mg_2_Ni. All of these results were consistent with those of the SEM and shear strength test. 

## 4. Discussion

The joint formation process in this work is shown in Figure 9. Figure 9a shows the sample before ultrasonic treatment. MgO layers are formed on the Mg BM as the BM was heated in air. After the USV is turned on, the MgO layers are broken rapidly, as shown in Figure 9b. This breakdown of the oxide layer results from the plasticity mismatch between the oxide layers and the Mg BM, which is caused by the acoustic softening effect [34,35]. Acoustic softening is the reduction in apparent static stress necessary for plastic deformation in a material under the influence of ultrasonic energy. Sayed et al. pointed out that the reduction of stress by acoustic softening was considered to be proportional to the acoustic intensity [39,40]. While Huang et al. discovered that the stress reduction was proportional to the vibration amplitude [41]. A model of the acoustic softening effect based on the thermal activation theory was cited by Yao et al. [42,43]. It is reported that applying high-frequency vibration during the plastic deformation leads to a stress reduction, which is called the ultrasonic volume effect [35]. The stress reduction τ can be expressed as Equation (2) [44,45]
(2)$$\tau =\hat{\tau }[1-kT\ln\left(\dot{\gamma_{0}}/\dot{\gamma_{p}}\right)/\Delta F,]$$
where *k* is the Boltzmann constant; *T* is the Kelvin temperature; $\dot{\gamma_{0}}$ is the pre-exponential factor, which is a constant between −1 and 1 and is obtained by the experiment;$\dot{\;\gamma_{p}}$ is the shear plastic strain rate obtained by the experiment; ΔF is the activation energy required to overcome the obstacle without the help from external stress. $\hat{\tau }$ is the mechanical threshold, which is usually expressed as Equation (3) [46]
(3)$$\hat{\tau }=\tau_{0}+\mu \alpha b\sqrt{\rho },$$
where $\tau_{0}$ is the friction stress; *μ* is the elastic shear modulus; *α* is a coefficient close to 1/3; *b* is the length of the Burgers vector; *p* is the dislocation density. Replacing $\;kT\ln\left(\dot{\gamma_{0}}/\dot{\gamma_{p}}\right)/\Delta F$ with *W*, Equation (2) changes to Equation (4)
(4)$$\tau =\hat{\tau }\left(1-W\right).$$

Therefore, in addition to the acoustic intensity and vibration amplitude, the stress reduction *τ* also depends on some physical properties of the materials such as $\tau_{0}$, *μ*, ΔF, and *b*, which are listed in Table 2. Based on the stress-reduction equation, the difference of the acoustic softening effect during the USV process between the Mg alloy and its oxide layer will lead to their plasticity mismatch, which eventually causes the breakdown of the surface oxide layer. 

After the MgO layer was broken in some areas, the fresh Mg and Ni atoms come to contact with each other and the Mg_2_Ni IMC layer generated by mutual diffusion is formed at the surface of the Ni interlayer within 10 s ultrasonic duration time, which is shown in Figure 9c. It took 5 min to form this solid-state diffusion layer at 515 °C using TLP bonding for an AZ31/Ni/AZ31 joint [38]. So, it can be concluded that the diffusion time between the Ni interlayer and the Mg BM is greatly shortened by the ultrasonic treatment. This is mainly due to the fact that the acoustic pressure affects the mobility, the diffusion potential, and the boundary conditions for diffusion [55,56,57]. The stress as a driving force for diffusion in formation of solute-atom atmosphere around dislocations was studied [58]. In a system containing a stress field, a diffusing particle generally experiences a force in a direction that reduces its interaction energy with the stress field. To find the driving force exerted on an interstitial solute atom by the stress field, the entropy production must be considered. In a small cell embedded in the material, the energy caused by stress-induced diffusion is presented as Equation (5)
(5)$$dw=-P\Delta \Omega_{1}dc_{1},$$
where $\Delta \Omega_{1}$ is a pure dilation caused by the interstitial space; *P* is pressure; $c_{1}$ is the concentration of atoms. For the isotropic elastic material, $P=-\frac{\sigma_{xx}+\sigma_{yy}+\sigma_{zz}}{3}$. The total driving force, including chemical potential, can be represented as Equation (6)
(6)$$\vec{F_{1}}=-\nabla \left(\mu_{1}+\Delta \Omega_{1}P\right).$$

The diffusion potential is an ‘elastochemical’ type of potential corresponding to Equation (7)
(7)$$\Phi_{1}=\mu_{1}+\Delta \Omega_{1}P.$$

Therefore, the entropy flux is Equation (8)
(8)$$\vec{J_{1}}=L_{11}{\vec{F}}_{1}=-L_{11}\nabla \Phi_{1}=-D_{1}\left(\nabla c_{1}+\frac{c_{1}\Delta \Omega_{1}}{kT}\nabla P\right),$$
where *L*_11_ is the Onsager coupling coefficient that can be calculated by introducing diffusivity *D*_1_.

Based on Equations (5)–(8), the relationship between the entropy flux $\vec{J_{1}}$ and driving force $\;{\vec{F}}_{1}$ has been revealed qualitatively, which indicates that the entropy flux to overcome the block of dilation decreases and the atom diffusion in solids will be accelerated with the additional driving force from ultrasonic vibration. 

So far, it can be seen that stress induces and influences diffusion directly. The quantitative analysis on mutual diffusion between Ni foil and Mg BMs with USV was conducted through DICTRA software (Pittsburgh, PA, USA). The pressure of TLP bonding in an AZ31/Ni/AZ31 joint is 0.0053 Pa [38], whereas in Liu’s research, the maximum pressure on the Al substrate is 6.08 × 10^5^ Pa with USV [59], which indicates that the pressure with USV is about 10^8^ times of that without USV. This intensified pressure was converted to an enhancement factor to apply in the simulation model. Figure 10 shows the simulation model and results of the mutual diffusion between the Ni foil and the Mg BM with, or without, USV from the DICTRA software. The temperature of this diffusion simulation is 520 °C. As the Mg_2_Ni IMC layer is formed at the interface between the Ni foil and the Mg BM, it is assumed that the concentration of Mg atoms is 0 at % along the centerline of the Ni foil and 0.67 at % at the reaction interface of the Ni foil side, meanwhile, the concentration of Ni atoms is 0at % at 200 μm away from the reaction interface in the Mg BM and 0.33 at % at the reaction interface of the Mg BM side. This composition boundary condition is shown in Figure 10a. It can be seen in Figure 10b,c that the diffusion distance of Ni atoms in the Mg BM is longer than that of Mg atoms in the Ni foil, which means that the Mg BM is consumed faster than the Ni foil, and the reaction interface mainly moves toward to the Mg BM side. Comparing atom concentration with and without USV, the concentration distribution of solute atoms are more uneven with USV than without USV. It can be seen in Figure 10c that the Ni composition in the Mg BM is inclined to increase and concentrate in some areas with increasing USV duration. That is, the diffusion distance of Ni atoms in the Mg BM that reach 0.33 at % is 14 μm when heated for 5 s without USV, whereas the diffusion distance of Ni atoms that reach 0.33 at % is 27 μm when heated for 5 s with USV. The diffusion distance of Ni atoms that reach 0.33 at % is 43 μm when heated for 30 s without USV, whereas the diffusion distance of Ni atoms that reach 0.33 at % is 136 μm when heated for 30 s with USV. Therefore, it can be concluded that the diffusion between the Mg BM and the Ni foil is accelerated by the introduction of USV. 

With the continuity of mutual diffusion and dissolution, the liquid eutectic phases composed of α-Mg and Mg_2_Ni will be generated, as shown in Figure 9d. Once the liquid phase is formed, the cavitation effect will lift the MgO layer, which is called undermining [60]. As the cavitation threshold of the liquids is 0.3–0.6 × 10^5^ Pa, which is much lower than the acoustic pressure caused by USV, a large number of cavitation bubbles are generated in the eutectic and the MgO layers are effectively removed by these bubbles [61]. In addition, due to the numerous bubbles that are ceaselessly formed and collapsed, the dissolution activation energy is decreased and the dissolution rate at the interface is accelerated remarkably [59,62]. Therefore, both the diffusion and the dissolution between the Ni foil and the Mg BM can be accelerated by USV and the Ni foil will be consumed within 27 s, which is shown in Figure 9e. 

In the period of 29–30 s ultrasonic treatment time, shown in Figure 9f, the width of the eutectic declines dramatically due to the acoustic steaming effect of USV. The flowing rate of the eutectic is accelerated and then a mass of eutectic is squeezed out [19,20,21]. The remaining structure is mainly composed of α-Mg (Ni). In other words, the reason for the increase of the shear strength to 107 MPa is that the brittle structure of the eutectic composed of Mg_2_Ni+α-Mg is squeezed out of the joints by the acoustic steaming effect, and the plastic and ductile structure of α-Mg (Ni) remains.

By far, the role that USV played in TLP bonding of Mg/Ni/Mg joints is summed up as follows. The MgO film on the Mg BM is broken by the acoustic softening effect and removed by undermining from the cavitation effect; the solid-state diffusion between the Ni foil and the Mg BM is accelerated by acoustic pressure, and the dissolution between the Ni foil and the Mg BM is increased by the cavitation effect; the brittle eutectic is extruded by the acoustic steaming effect, which increases the shear strength of the joints.

## 5. Conclusions

(1) Magnesium alloys can be joined with a Ni interlayer via U-TLP within 30 s of ultrasound treatment in air.

(2) The microstructure of the joints evolved from the Mg_2_Ni layer, the eutectic composed of Mg_2_Ni and α-Mg, to the α-Mg (Ni) solid solution with the increase of ultrasound treatment time.

(3) With incremental increases in ultrasonic duration, the shear strength increased from 19 MPa to the maximum value of 107 MPa; however, it declined to 35 MPa when the ultrasonic treatment time was prolonged to 35 s. If the continuous Mg_2_Ni layer along the Ni interlayer existed, the fracture path was prone to be along the Mg_2_Ni layer as the Mg_2_Ni is the hardest phase in the brazing seam. When the Ni interlayer was consumed, the fracture path originated and propagated along the eutectic. When the majority of Mg_2_Ni was squeezed out and α-Mg(Ni) remained, the fracture paths originated and propagated either in the Mg BM or in the brazing seam as the hardness of α-Mg(Ni) is close to that of the BM.

(4) The oxide film was broken by the acoustic softening effect at the stage where the Mg BM and Ni interlayer were in a solid state. The MgO film and Mg BM undergo different degrees of stress reduction as properties such as *τ**_0_*, *μ*, ΔF, and *b* of MgO and Mg are different when USV is occurring, which induces the plastic mismatch in deformation and helps to break the MgO film. The composition and diffusion distance of solute atom in matrix was increased by the acoustic pressure from USV, which helps to shorten the mutual diffusion time between the BM and the interlayer to less than 10 s. 

## Figures and Tables

**Figure 1 materials-12-03732-f001:**
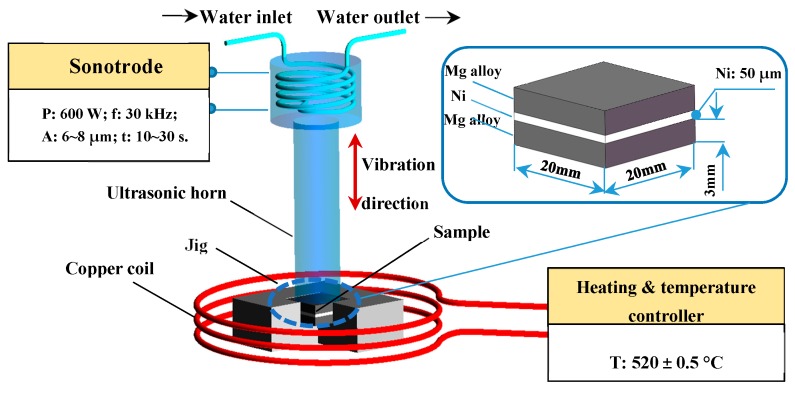
Schematic of ultrasound-assisted transient liquid phase bonding (U-TLP).

**Figure 2 materials-12-03732-f002:**
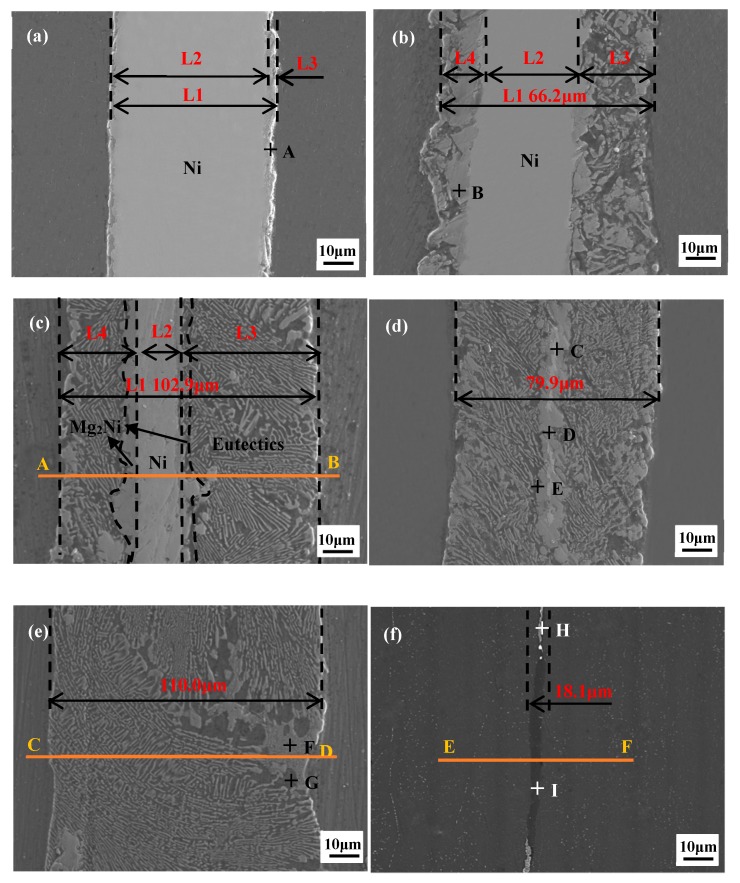
Microstructure of the joints with 10 s (**a**), 15 s (**b**), 20 s (**c**), 25 s (**d**), 27 s (**e**), and 30 s (**f**) ultrasonic treatment time.

**Figure 3 materials-12-03732-f003:**
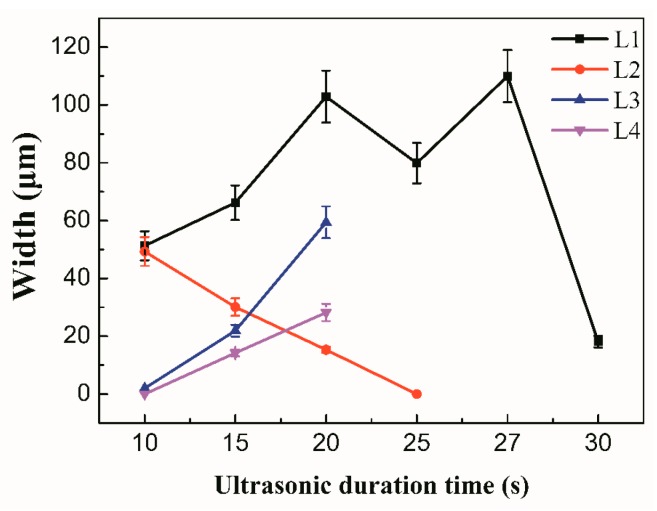
The width of zones L1–L4.

**Figure 4 materials-12-03732-f004:**
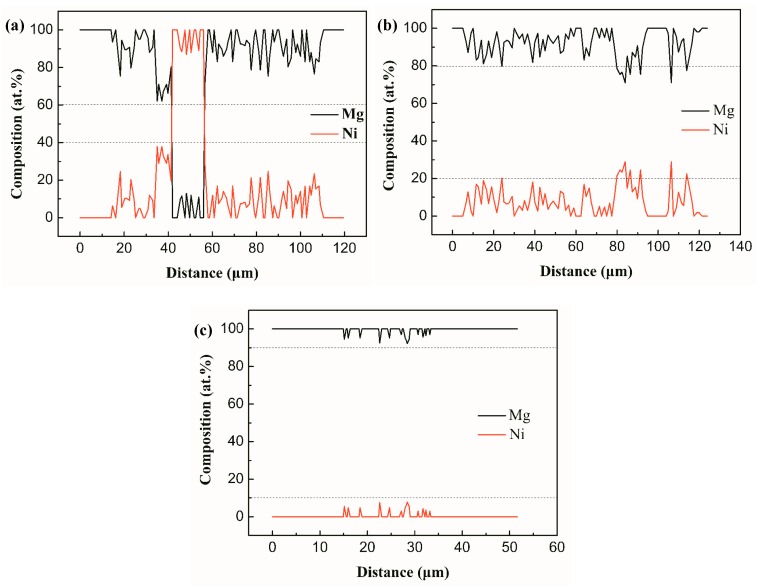
Distribution of elements across the joints (**a**) along line AB; (**b**) along line CD; (**c**) along line EF.

**Figure 5 materials-12-03732-f005:**
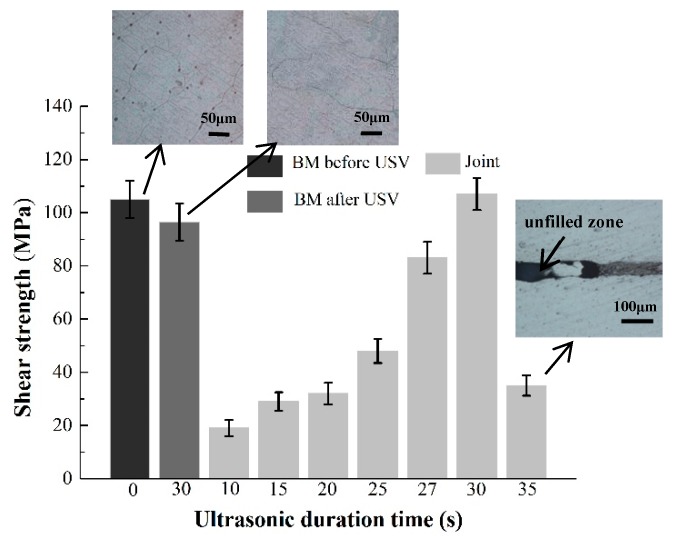
The shear strength of three samples of the base metals (BMs) and the joints at the given set of parameters.

**Figure 6 materials-12-03732-f006:**
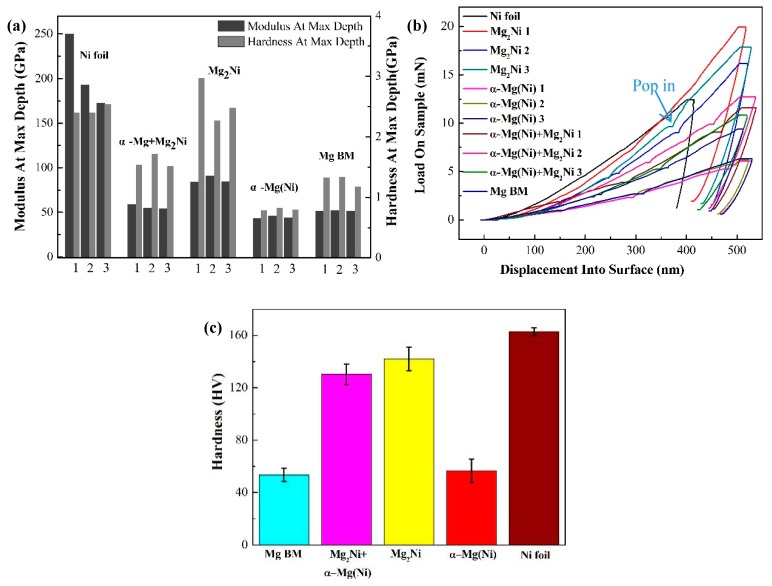
Nanoindentation hardness (**a**) hardness and modulus; (**b**) load–displacement curves, and (**c**) Vickers hardness at the microstructure.

**Figure 7 materials-12-03732-f007:**
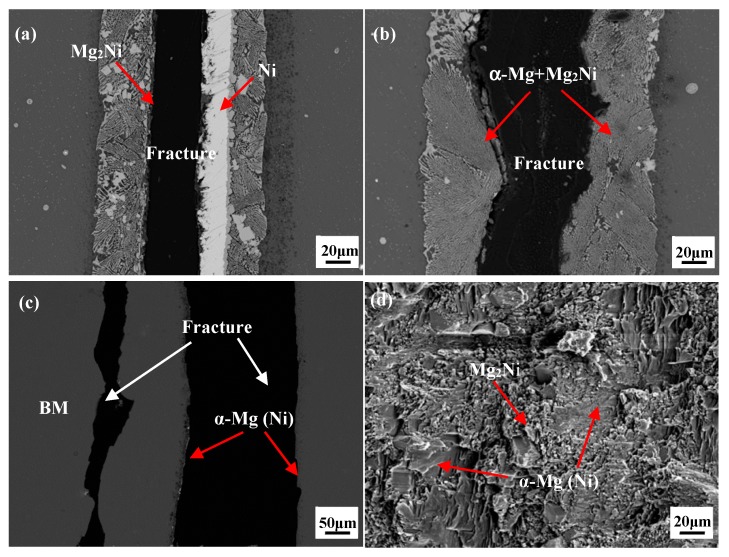
The fracture path in a series of ultrasonic treatment times (**a**) 20 s; (**b**) 27 s; (**c**) 30 s, and (**d**) morphology of the joint with 30 s ultrasonic treatment time.

**Figure 8 materials-12-03732-f008:**
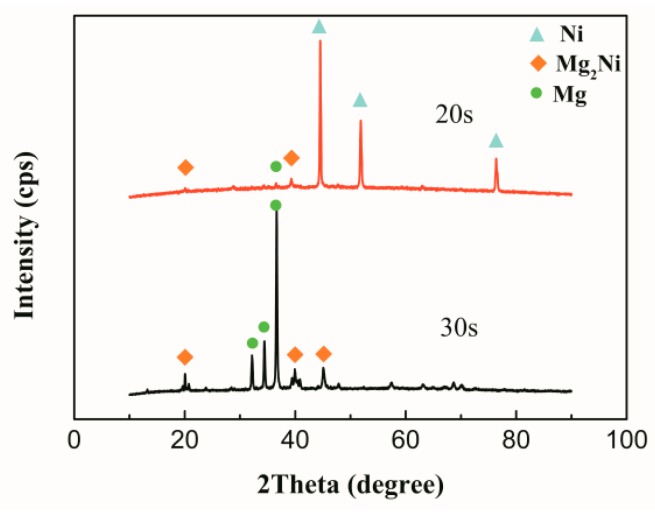
XRD analysis of fracture morphology.

**Figure 9 materials-12-03732-f009:**
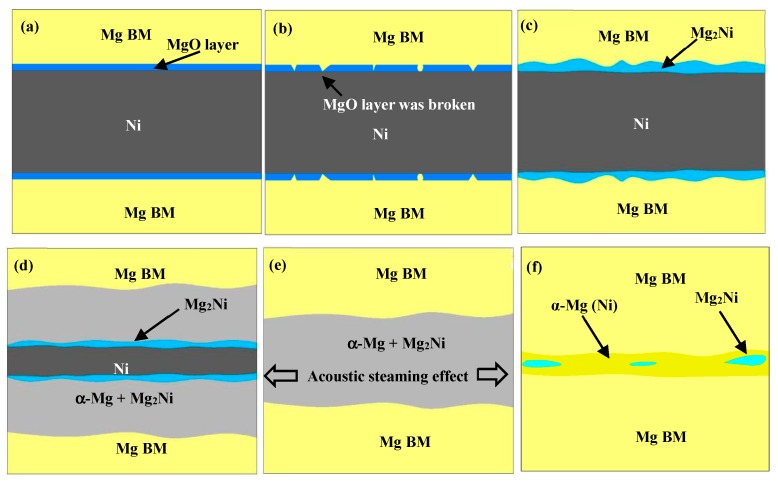
The joint formation mechanism (**a**) before ultrasonic treatment, (**b**) 0~9 s, (**c**) 10 s, (**d**) 11~26 s, (**e**) 27~29 s, (**f**) 30 s.

**Figure 10 materials-12-03732-f010:**
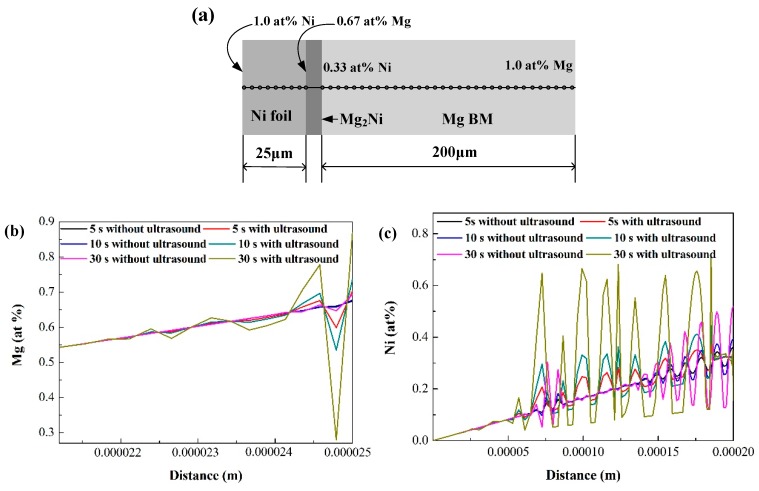
The schematic of solid-state diffusion (**a**), the diffusion results in the Ni interlayer (**b**) and the Mg BM (**c**) from DICTRA software calculation.

**Table 1 materials-12-03732-t001:** Chemical composition at different regions in Figure 2.

Point	Mg (at %)	Ni (at %)	Possible Phase
A	68.23	31.77	Mg_2_Ni
B	66.89	33.11	Mg_2_Ni
C	4.10	95.90	Ni
D	66.92	33.08	Mg_2_Ni
E	98.85	1.15	α-Mg(Ni)
F	67.04	32.96	Mg_2_Ni
G	97.80	2.20	α-Mg(Ni)
H	78.76	21.24	Mg_2_Ni + α-Mg(Ni)
I	98.90	1.10	α-Mg(Ni)

**Table 2 materials-12-03732-t002:** Property values of Mg alloys and MgO.

Parameters	Mg Alloys	MgO
Δ*F* (kJ/mol)	24.2 [47]	22.5 [48]
*b* (nm)	$1/3〈1\;1\;\bar{2}\;3〉$ [49]	$1/2〈1\;1\;\bar{0}〉$ [50]
*μ* (GPa)	31.2 [51]	16.1 [52]
$\tau_{0}$ (MPa)	30~145 [53]	40~55 [54]

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
