# Peer review of "Microstructure and Formation Mechanism of Ultrasound-Assisted Transient Liquid Phase Bonded Magnesium Alloys with Ni Interlayer"

_materials, 2019, doi:10.3390/ma12223732_

Round 1

Reviewer 1 Report

The first three sections of the article are well written and clear. The authors should just put the caption of Figure 5 to the same page as the picture and reveal the number of tested specimens for this figure.

The section 4 (Discussion), however, needs a major improvement. The sentences in lines 207-209 are unclear. In Table 2 there is no data on gamma_0 (rate), gamma_p (rate) and ro. Figure 9 is divided across two pages, which should not happen. It should be completely in one page. Why did the authors put equations (3) to (6) into the article? What is the meaning of c1 parameter? Were these equations used in DICTRA software? What were the settings of the calculations, which results are presented in Figure 10? I think that this questions need to be answered for the section to be more clear.

Author Response

Q: The first three sections of the article are well written and clear. The authors should just put the caption of Figure 5 to the same page as the picture and reveal the number of tested specimens for this figure.

A: The caption and the picture of figure 5 were put into the same page. And caption shows the number of the tested samples.

Q: The section 4 (Discussion), however, needs a major improvement. The sentences in lines 207-209 are unclear.

A: It has been corrected.

Q: In Table 2 there is no data on gamma_0 (rate), gamma_p (rate) and ro.

A: gamma_0 (rate) is a constant between -1 and 1.  gamma_p (rate) is variable changed with the temperature. Both of these two parameters can be obtained by certain experiments. However, it is hard for us to do these tests due to the restriction of the experimental equipment and conditions. Therefore, the  is replaced by W according to some published papers and stress reduction t depends on some physical properties of materials such as , m, and b, which are listed in the Tab.2.The revision of this part is listed in lines 220-228.

Q: Figure 9 is divided across two pages, which should not happen. It should be completely in one page.

A: Fig.9 was put into one page.

Q: Why did the authors put equations (3) to (6) into the article? What is the meaning of c1 parameter? Were these equations used in DICTRA software?

A: This is good question. These equations are used to reveal the qualitative relationship between the acoustic pressure that acts as driving force and the energy that the atom diffusion needs.  And the explanation of these equations has been added to the text in lines 246-260.

c1 is the concentration of atom, which is added to the text in line 248.

And these equations were not used in DICTRA software.

Q: What were the settings of the calculations, which results are presented in Figure 10?

A: The calculation setting and results have been revised, which are listed in lines 261-284.

Reviewer 2 Report

Overall good paper detailing methods and results. A few minor corrections to consider 

Some figures are like 2f or 7e are too dark to see. Improve the brightness  Some labels on the figures are cut-off in half. Please fix them.  Page 2 - experimental - Be more specific about the roughness intended at the interface. 800 grit size may not be smooth or rough enough. Why 800? Why not 4000? Do you clean the foil and the BM with acids/solvents? If not why? Does that help remove oxide layers? What is the temperature drop rate and after how long in the process you remove samples? What is the effect of keeping samples in the furnace after the ultrasound waves are turned off.  What happens to Mn and Ce impurities in the BM? Are you sure they do not interact with Ni? 

Author Response

Overall good paper detailing methods and results. A few minor corrections to consider 

Q: Some figures are like 2f or 7e are too dark to see. Improve the brightness Some labels on the figures are cut-off in half. Please fix them. 

A: The Figures have been improved according to the reviews.

Q: Page 2 - experimental - Be more specific about the roughness intended at the interface. 800 grit size may not be smooth or rough enough. Why 800? Why not 4000?

A: Based on previous experiment and published papers, the BMs were grounded by SiC papers to 800 grit finish to keep the surface roughness as 12.5mm, which helps to remove the oxide film during U-TLP bonding. And the revision is listed in lines 53-54.

Q: Do you clean the foil and the BM with acids/solvents? If not why? Does that help remove oxide layers?

A: It is unnecessary to clean the BMs and the foil with acids/solvents. This is mainly because that the residual oxide film or the new oxide film during heating will be removed by the function of ultrasonic vibration, which is the one of advantages of U-TLP method. And this function of removing oxide film has been researched and discussed in some previous papers.

Q: What is the temperature drop rate and after how long in the process you remove samples? What is the effect of keeping samples in the furnace after the ultrasound waves are turned off. 

A: The temperature drop rate is about 50 °C/min and the samples were removed when the temperature of the samples were cooled down below the 200 °C. Keeping the samples with a constant preload pressure above 200 °C helps to improve the bonding during the solidification. And the revision is listed in lines 59-62.

Q: What happens to Mn and Ce impurities in the BM? Are you sure they do not interact with Ni?  

A: This is a good question that I have thought about it since I proceeded these series of experiments. To identify the element and its composition, each phase was tested more than 5 times by EDS analysis. And the EDS results show that the Intermetallic compounds are composed of elements Mg and Ni and the a-Mg solutions are composed of Mg atoms and a minority of Ni atom. Therefore, the element Mn and Ce are not participated the reaction under these sets of experiments.

Round 2

Reviewer 1 Report

My comments were adequately considered.